

# Solving the clustered traveling salesman problem *via* traveling salesman problem methods

Yongliang Lu[1], Jin-Kao Hao[2] and Qinghua Wu[3]

[1] School of Economics and Management, Fuzhou University, Fuzhou, China
[2] LERIA, Université d'Angers, Angers, France
[3] School of Management, Huazhong University of Science and Technology, Wuhan, China

## ABSTRACT

The Clustered Traveling Salesman Problem (CTSP) is a variant of the popular Traveling Salesman Problem (TSP) arising from a number of real-life applications. In this work, we explore a transformation approach that solves the CTSP by converting it to the well-studied TSP. For this purpose, we first investigate a technique to convert a CTSP instance to a TSP and then apply powerful TSP solvers (including exact and heuristic solvers) to solve the resulting TSP instance. We want to answer the following questions: How do state-of-the-art TSP solvers perform on clustered instances converted from the CTSP? Do state-of-the-art TSP solvers compete well with the best performing methods specifically designed for the CTSP? For this purpose, we present intensive computational experiments on various benchmark instances to draw conclusions.

# INTRODUCTION

The Clustered Traveling Salesman Problem (CTSP), originally proposed by *Chisman (1975)*, is an extension of the classic Traveling Salesman Problem (TSP) where the cities are grouped into clusters and the cities of each cluster must be visited contiguously. Formally, the problem is defined on a symmetric complete weighted graph $G = (V, E)$ with a set of vertices $V = \{1,2,\dots,n\}$ and a set of edges $E = \{(i, j):i, j \in V, i \neq j\}$. The vertex set $V$ is partitioned into disjoint clusters $V_1,V_2,\dots,V_m$ ($V_1 \cup V_2 \cup\dots\cup V_m = V$). Let $C$ be an $n \times n$ symmetric distance matrix such that $c_{ij}$ ($i, j = 1,2\dots,n, i \neq j$) represents the travel cost between two corresponding vertices $i$ and $j$, and satisfies the triangle inequality rule. The objective of the CTSP is to find a minimum cost Hamiltonian circuit over all the vertices, where the vertices of each cluster must be visited consecutively.

Figure 1 shows a feasible solution for a CTSP instance, where the solution corresponds to a Hamiltonian cycle such that the vertices of each cluster are visited contiguously.

The CTSP can be formally modelled as the following integer programming model described in *Chisman (1975)* where, without loss of generality, the salesman is assumed to leave origin city 1 and return to 1.

Corresponding author
Qinghua Wu,
qinghuawu1005@gmail.com

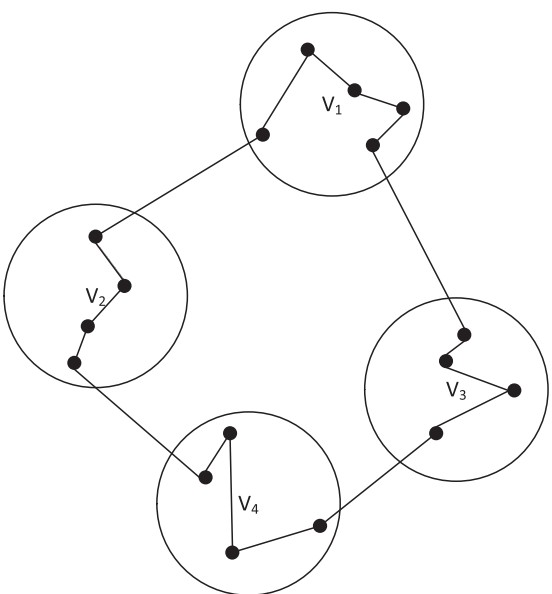

**Figure 1 A feasible solution for an instance of the CTSP.**

$$\min f = \sum_{i=1}^{n}\sum_{j=1}^{n} c_{ij}x_{ij} \tag{1}$$

subject to

$$\sum_{j=1}^{n} x_{ij} = 1 \qquad \forall i \in V \tag{2}$$

$$\sum_{i=1}^{n} x_{ij} = 1 \qquad \forall j \in V \tag{3}$$

$$u_i - u_j + (n-1)x_{ij} \leq n-2 \qquad 2 \leq i \neq j \leq n \tag{4}$$

$$\sum_{i \in V_k}\sum_{j \in V_k} x_{ij} = |V_k| - 1 \qquad \forall V_k \subset V, |V_k| \geq 1, k = 1, 2, \ldots, m \tag{5}$$

$$x_{ij} \in \{0, 1\} \qquad \forall i, j \in V \tag{6}$$

$$u_i \geq 0 \qquad 2 \leq i \leq n \tag{7}$$

In this model, the binary variable $x_{ij} = 1$ if city $j$ is visited immediately after city $i$; $x_{ij} = 0$ otherwise. Objective function (1) seeks to minimize the total distance traveled by the salesman. Constraints (2) and (3) ensure that each city is visited exactly once. Constraints (4) eliminate subtours, while constraints (5) guarantee that the cities of each cluster are visited contiguously. The remaining constraints are related to the decision variables.

The above subtour elimination constraints (4) are called MTZ formulation (*Miller, Tucker & Zemlin, 1960*). Although MTZ is simple to implement, it provides a very poor linear relaxation (*Campuzano, Obreque & Aguayo, 2020*). Many compact formulations have been proposed to replace Constraints (4). According to the literature,

a multi-commodity flow formulation (*Wong, 1980*; *Claus, 1984*) was proven to provide a strong linear relaxation, without compromising its simplicity. In the multi-commodity flow formulation, let $k = 2,3,\ldots,n$ be $n-1$ commodities, and let $y_{ij}^k$ be a nonnegative decision variable which represents the flow on the arc $(i, j) \in E$ for the commodity $k$ from city 1 to city $k$. Then, another alternative mathematical model for the CTSP is constituted of the objective function (1) and the constraints (2), (3), (5), (6) along with the following subtour elimination constraints:

$$0 \leq y_{ij}^k \leq x_{ij} \qquad \forall i,j,k \in V, k \neq 1 \tag{8}$$

$$\sum_{i=2}^{n} y_{1i}^k = 1 \qquad \forall k \in V \setminus \{1\} \tag{9}$$

$$\sum_{i=2}^{n} y_{i1}^k = 0 \qquad \forall k \in V \setminus \{1\} \tag{10}$$

$$\sum_{i=1}^{n} y_{ik}^k = 1 \qquad \forall k \in V \setminus \{1\} \tag{11}$$

$$\sum_{j=1}^{n} y_{kj}^k = 0 \qquad \forall k \in V \setminus \{1\} \tag{12}$$

$$\sum_{i=1}^{n} y_{ij}^k - \sum_{i=1}^{n} y_{ji}^k = 0 \qquad \forall j,k \in V \setminus \{1\}, j \neq k \tag{13}$$

Constraints (8) only allow flow in an arc $(i, j)$ if and only if it is traversed by the salesman (*i.e.*, $x_{ij} = 1$). Constraints (9) ensure that city 1 is the source of one unit of each commodity $k \in V \setminus \{1\}$ and Constraints (10) avoid that the flow of each commodity $k \in V \setminus \{1\}$ returns to city 1. Constraints (11) and (12) guarantees that one flow unit of commodity $k$ enters to city $k$ and it does not leave the city $k$. Constraints (13) ensure flow conservation at each city, apart from city 1 and for commodity $k$ at city $k$.

One notices that the CTSP is equivalent to the TSP when there is a single cluster or when each cluster contains exactly one vertex. Therefore, the CTSP is NP-hard, and thus computationally challenging in the general case. From a practical perspective, the CTSP is a versatile modeling tool for several operational research applications arising in a wide variety of areas, including automated warehouse routing (*Chisman, 1975*), emergency vehicle dispatching (*Weintraub et al., 1999*), production planning (*Lokin, 1979*), disk defragmentation (*Laporte & Palekar, 2002*), and commercial transactions with supermarkets, shops and grocery suppliers (*Ghaziri & Osman, 2003*). As a result, effective solution methods for the CTSP can help to solve these practical problems. Indeed, the computational challenge and the wide range of applications of the problem have motivated a variety of approaches that are reviewed in the "Literature Review on Existing Solution Methods" section. However, unlike the classic TSP problem for which many powerful methods have been introduced in the past decades, studies on the CTSP are still quite limited.

Moreover, the CTSP belongs to the large class of traveling salesman problems. Among the TSP variants, the generalized traveling salesman problem (GTSP) (*Srivastava et al., 1969*; *Cosma, Pop & Cosma, 2021*) and the family traveling salesman problem (FTSP) (*Morán-Mirabal, Velarde & Resende, 2014*; *Pop, Matei & Pintea, 2018*) share similarities with the CTSP. In the GTSP, the set of vertices is divided into clusters and the objective is to find a minimum-cost tour passing through one vertex from each cluster. In the FTSP, the set of vertices is also divided into clusters (called families) and the objective is to visit a predefined number of vertices in each family at a minimum cost.

In this work, we investigate the problem transformation approach proposed in *Chisman (1975)*, which converts the CTSP to the TSP and assess the interest of popular modern TSP solvers for solving the resulting TSP instances. To our knowledge, this is the first large computational study testing modern TSP solvers on solving the CTSP. The work is motivated by the following considerations. First, intensive researches on the TSP have led to the development of many very powerful solvers. Thus, it is interesting to know whether we can take advantage of these solvers to effectively solve the CTSP. Second, the TSP instances converted from the CTSP are characterized by their cluster structures. These instances constitute interesting test cases for existing TSP solvers. This work aims thus to answer the following questions.

1. How do state-of-the-art *exact* TSP solvers perform on clustered instances converted from the CTSP?

2. How do state-of-the-art *inexact* (heuristic) TSP solvers perform on clustered instances converted from the CTSP?

3. Do state-of-the-art TSP solvers compete well with the best performing methods specifically designed for the CTSP?

To our knowledge, Questions 1 and 3 have never been investigated in the literature. Regarding Question 2, two previous studies (*Neto, 1999*; *Helsgaun, 2014*) are of interest. However, they are limited because they only concern one TSP algorithm, *i.e.*, the local search based LKH solver (*Helsgaun, 2009*), while ignoring other powerful TSP solvers like GA-EAX (*Nagata & Kobayashi, 1997*) and Concorde (*Applegate, Bixby & Chvatal, 2006*). Answering these questions helps to enrich the state-of-the-art of solving the CTSP and gain novel knowledge on using modern TSP methods to solve new problems. Finally, we mention that the transformation approach was also tested in *Lokin (1979)* and *Jongens & Volgenant (1985)*. However, these studies are clearly outdated and don't provide useful information as to the questions we want to investigate.

The remainder of this paper is organized as follows. "Literature Review on Existing Solution Methods" reviews existing solution methods for the CTSP. "Solving the CTSP *via* TSP Methods" presents the transformation of the CTSP to the TSP and three powerful TSP methods (solvers). "Computational Experiments" shows computational studies of the TSP solvers applied to the clustered instances and comparisons with existing algorithms dedicated to the CTSP. "Discussion" provides additional explanations regarding the

behaviors of the three TSP solvers. Finally, concluding remarks are provided in "Conclusion".

## LITERATURE REVIEW ON EXISTING SOLUTION METHODS

There are several dedicated solution algorithms for solving the CTSP that are based on exact, approximation, and metaheuristic approaches.

Along with the introduction of the CTSP, *Chisman (1975)* proposed a branch-and-bound algorithm to solve the integer programming model presented in the Introduction section. *Jongens & Volgenant (1985)* developed an algorithm based on the 1-tree relaxation to provide lower bounds as well as a heuristic to find satisfactory upper bounds. *Mestria, Ochi & de Lima Martins (2013)* used the mathematical formulation of *Chisman (1975)* and IBM Parallel CPLEX solver (version 11.2) to obtain lower bounds for medium CTSP instances ($|V| \leq 1,000$).

Various $a$-approximation algorithms (*Anily, Bramel & Hertz, 1999*; *Gendreau, Laporte & Hertz, 1997*; *Guttmann-Beck et al., 2000*) have been developed for the CTSP. These approximation algorithms require either the starting and ending vertices in each cluster or a prespecified order of visiting the clusters in the tour as inputs, and solve the inter-cluster and intra-cluster problems independently. *Bao & Liu (2012)* presented a new 2.17-approximation algorithm where no starting and ending vertices were specified. Later, *Bao et al. (2017)* provided a 2.5-approximation algorithm for another version of the CTSP where the starting vertex of each cluster is given while the ending vertex is not specified. Recently, *Kawasaki & Takazawa (2020)* improved the approximation ratio for the CTSP by incorporating a recent approximation algorithm for the TSP by *Zenklusen (2019)*.

Given that the CTSP is a NP-hard problem, a number of heuristic and metaheuristic algorithms have also been investigated, which aim to provide high-quality solutions in acceptable computation time, but without provable optimal guarantee of the attained solutions. For example, *Laporte, Potvin & Quilleret (1997)* presented a tabu search algorithm to solve a particular case of the CTSP, where the clusters are visited in a prespecified order. *Potvin & Guertin (1996)* developed a genetic algorithm for the CTSP that finds inter-cluster paths and then intra-cluster paths. Later, *Ding, Cheng & He (2007)* proposed a two-level genetic algorithm for the CTSP. In the first level, a genetic algorithm is used to find the shortest Hamiltonian cycle for each cluster. In the second level, a modified genetic algorithm is applied to merge the Hamiltonian cycles of all the clusters into a complete tour.

In addition to these early heuristic algorithms, *Mestria, Ochi & de Lima Martins (2013)* investigated GRASP (Greedy Randomized Adaptive Search Procedure) with path-relinking. Among the six proposed heuristics, one heuristic corresponds to the traditional GRASP procedure whereas the other heuristics include different path relinking procedures. *Mestria (2016)* studied a hybrid heuristic, which is based on a combination of GRASP, Iterated Local Search (ILS) and Variable Neighborhood Descent (VND). Recently, *Mestria (2018)* presented another complex hybrid algorithm (VNRDGILS) which mixes GRASP, ILS, and Variable Neighborhood Random Descent to explore several neighborhoods. According to the computational results reported in *Mestria, Ochi & de*

*Lima Martins (2013)* and *Mestria (2016, 2018)*, these GRASP-based algorithms are among the best performing heuristics specially designed for the CTSP currently available in the literature. In addition, *Hà et al. (2022)* proposed a metaheuristic method based on the ILS framework with problem-tailored operators for a version of the CTSP where the order of visiting the clusters is prespecified.

Existing studies have significantly contributed to better solving the CTSP. According to the computational results reported in the literature, due to the NP-hardness of the problem, only small CTSP instances were able to be solved to optimality with the exact algorithms. The approximation algorithms provide solutions for the CTSP within a given approximation factor. However, due to the high approximation factors involved (*e.g.*, 5/3 (*Anily, Bramel & Hertz, 1999*), 3/2 *Gendreau, Laporte & Hertz (1997)*, 2.17 (*Bao & Liu, 2012*), and 2.5 (*Bao et al., 2017*)), these approximation algorithms are not practical for solving large instances. To deal with large CTSP instances, heuristic and metaheuristic algorithms are often preferred to find sub-optimal solutions within an acceptable computation time.

## SOLVING THE CTSP *VIA* TSP METHODS

### Transformation of the CTSP to the TSP

As the literature review shows, a number of dedicated solution approaches have been developed to solve the CTSP. However, one observes that these approaches have difficulty producing robustly and consistently high-quality solutions for large-scale CTSP instances with tens of thousands of vertices. Moreover, the best performing CTSP methods (*e.g.*, VNRDGILS (*Mestria, 2018*), HHGILS (*Mestria, 2016*), and GPR1R2 (*Mestria, Ochi & de Lima Martins, 2013*)) are computationally expensive (*e.g.*, requiring 1,080 s to find good solutions for instances with $1,173 \leq n \leq 2,000$).

On the other hand, problem transformation has been highly successful in solving several difficult optimization problems such as the latin square completion problem *via* graph coloring (*Jin & Hao, 2019*) and the winner determination problem *via* weighted maximum cliques (*Wu & Hao, 2015*). It is known that the CTSP can be transformed to the conventional TSP (*Chisman, 1975*). Therefore, in principle, the CTSP can be solved by any TSP algorithm. However, to our knowledge, no computational study on using problem transformation to solve the CTSP has been presented in the literature. This work fills the gap by exploring the problem transformation approach of *Chisman (1975)* and testing three representative state-of-the-art TSP solvers including both exact and inexact solution approaches.

The basic idea of transforming the CTSP to the TSP is to add a large artificial cost $M$ to all inter-cluster edges in order to force the salesman to visit all the cities within each cluster before leaving it.

Given a CTSP instance $G = (V, E)$ with distance matrix $C$, we define a TSP instance $G' = (V', E')$ with distance matrix $C'$ as follow.

- Define $V = V'$ and $E = E'$.
- Define the travel distance $c'_{ij}$ in $G'$ by

$$c'_{ij} = \begin{cases} c_{ij} + M & \text{if } i \text{ and } j \text{ belong to different clusters} \\ c_{ij} & \text{otherwise} \end{cases}$$

Obviously, if the value of $M$ is sufficiently large, then the best Hamiltonian cycle in $G'$ is a feasible CTSP solution in $G$, in which the vertices of each cluster are visited contiguously.

**Property.** *An optimal solution to the TSP instance corresponds to an optimal solution to the original CTSP instance.*

**Proof.** Let $S'$ and $S$ be the optimal solutions of the TSP instance $G'$ and the original CTSP instance $G$, respectively. Let $m$ be the number of clusters of $G$. To minimize the total travel cost, there are only $m$ inter-cluster edges in $S'$. Therefore, $S'$ is a feasible CTSP solution for $G$ and satisfies the following relation:

$$f(S') = f(S) + m \times M$$

Obviously, $S'$ corresponds to $S$ by subtracting the constant $m \times M$.

## Solution methods for the TSP

There are numerous solution methods for the TSP. In this work, we adopt three very powerful TSP solvers whose codes are publicly available, including one exact solver (Concorde (*Applegate, Bixby & Chvatal, 2006*)) and two inexact (heuristic) solvers (LHK-2 (*Helsgaun, 2009*) and GA-EAX (*Nagata & Kobayashi, 2013*)).

Notice that the TSP instance converted from a CTSP instance has a particular feature that the vertices are grouped into clusters and the distance between each pair of vertices within a same cluster is in general small, while this distance is large for two vertices from different clusters. Along with the presentation of the TSP solvers, we discuss their suitability for solving such clustered instances each time this is appropriate.

### Exact Concorde solver

Concorde is an advanced exact TSP solver for the symmetric TSP based on Branch-and-Bound and problem specific cutting plane methods (*Applegate, Bixby & Chvatal, 2006*). It makes use of a specifically designed QSopt linear programming solver. According to *Hoos & Stützle (2014)*, Concorde is the best performing exact algorithm for the TSP. As shown in *Applegate et al. (2006)*, Concorde can solve benchmark instances from TSPLIB with up to 1,000 vertices to optimality within a reasonable computation time and it also solves large TSP instances at the cost of a long computation time.

The run time behavior of Concorde has been investigated essentially on random uniform instances. For instance, *Applegate et al. (2006)* investigated the run time required by Concorde for solving random uniform instances and indicated that the run time increases as an exponential function of instance size $|V|$. *Hoos & Stützle (2014)* further demonstrated that the median run time required by Concorde scales with instance size $|V|$ of the form $ab^{\sqrt{|V|}}$ ($a \approx 0.21$, $b \approx 1.24$) on the widely studied class of uniform random TSP instances. To our knowledge, no study has been reported concerning the behavior of

Concorde on sharply clustered instances. As a result, the current study will provide useful information on this issue.

### Lin-Kernighan based heuristic solver

According to the TSP literature, a majority of the best performing TSP heuristic algorithms is based on the Lin-Kernighan (LK) heuristic (*Lin & Kernighan, 1973*) and its extensions. The LK heuristic is a variable-depth $k$-opt local search procedure, where the $k$-opt neighborhood is partially searched with a smart pruning strategy. LK explores the most promising neighbors within the $k$-opt neighborhood, that is, the set of feasible tours obtained by removing $k$ edges and adding other $k$ edges such that the resulting tour is feasible. Several improved versions of the basic LK heuristic have been introduced within the iterated local search framework (*e.g.*, *Applegate, Cook & Rohe, 2003*; *Helsgaun, 2000*; *Helsgaun, 2009*; *Martin, Otto & Felten, 1991*).

Among these iterated LK algorithms, Helsgaun's LKH (*Helsgaun, 2000*, *2009*) is the uncontested state-of-the-art heuristic TSP solver. *Helsgaun (2000)* developed an iterated version of LK together with an efficient implementation of the LK algorithm, known as the Lin-Kernighan-Helsgaun (LKH-1) heuristic, where a 5-opt move is used as the basic move to broaden the search and an $\alpha$-measure method based on sensitivity analysis of minimum spanning trees is used to restrict the search to relative few of the $\alpha$-nearest neighbors of a vertex to speed up the search process. Later, *Helsgaun (2009)* further extended LKH-1 by developing a highly effective implementation of the $k$-opt procedure (called LKH-2), which eliminated many of the limitations and shortcomings of LKH-1. Furthermore, LKH-2 specially extended the data structures of LKH-1 to solve very large TSP instances. The main features of LKH-2 include (1) using sequential and non-sequential $k$-opt moves, (2) using several partitioning procedures to partition a large TSP instance into smaller subproblems, (3) using a tour merging procedure to generate a better solution from two or more local optimum solutions, and (4) applying a backbone-guided search to guide the local search to make biased local perturbations. LKH-2 is considered to be one of most effective heuristic methods for finding very high-quality solutions for various large TSP instances (*Dubois-Lacoste, Hoos & Stützle, 2015*).

However, the LK algorithm and any LK-based algorithms require much longer running times on clustered instances of the TSP than on uniformly distributed instances (*Neto, 1999*). The main reason why the LK heuristic stumbles on clustered instances is that relatively large inter-cluster edges serve as bait edges. During the LK search, when removing such a bait edge, the LK heuristic is tricked into long and often fruitless searches. More precisely, each time an edge bridging two clusters is removed, the cumulative gain rises enormously, and the procedure is encouraged to perform very deep searches. To alleviate the problem, a cluster compensation technique was proposed in *Neto (1999)* for the Lin-Kernighan heuristic to limit its performance degradation. *Helsgaun (2009)* showed that the LKH-2 algorithm performs significantly worse on sharply clustered instances than on uniform random instances. To remedy this difficulty, *Helsgaun (2014)* considered the unusual structure of clustered instances, and adjusted the parameter

---

**Algorithm 1** GA-EAX for the CTSP.

**Require:** TSP instance $G$, population size $p$; number of offspring solutions $r$ generated from each parent pair

**Ensure:** best solution $S^*$

1: $POP = \{P_1, P_2, ..., P_p\} \leftarrow$ Initial_Population($G$)

2: **while** stopping condition is not met **do**

3:      Randomly shuffle the solutions in $POP$

4:      **for** $i = 1,2,..., p$ **do**

5:          $S_1 \leftarrow P_i$, $S_2 \leftarrow P_{i+1}$ /* Note: $P_{p+1} = P_1$ */

6:          $(o_1, ..., o_r) \leftarrow$ EAX($S_1$, $S_2$)

7:          $P_i \leftarrow$ Select_Best($o_1,..., o_r$, $S_1$)

8:      **end for**

9: **end while**

10: $S^* \leftarrow$ Best($POP$)

11: Return $S^*$

---

settings of LKH-2 to better solve the clustered instances. The resulting solver is named CLKH, which is used in this study.

### Edge assembly crossover based genetic algorithm

Population-based evolutionary algorithms are another well-known approach for the TSP. A popular example is the powerful genetic algorithm introduced by *Nagata & Kobayashi (2013)*. This algorithm (called GA-EAX, see Algorithm 1) is characterized by its powerful edge assembly crossover (EAX) operator introduced in *Nagata & Kobayashi (1997)* and *Nagata & Soler (2012)* with an efficient implementation and a cost-effective selection strategy for maintaining population diversity.

The key EAX operator generates, from two high-quality tours (parents), one offspring tour by first inheriting the edges from the parents to construct disjoint subtours and then connecting the subtours with new edges in a greedy fashion (similar to building a minimal spanning tree). Let $S_A$ and $S_B$ be the parents, EAX operates as follows (see Fig. 2 for an example):

1. Generate an undirected multigraph defined as $G_{AB} = (V, E_A \cup E_B)$, where $E_A$ and $E_B$ are the sets of edges of parents $S_A$ and $S_B$, respectively.

2. Extract all AB-cycles from $G_{AB}$. An AB-cycle is defined as a cycle in $G_{AB}$, such that edges of $E_A$ and edges of $E_B$ are alternately linked.

3. Construct an E-set by selecting AB-cycles according to a given selection strategy (*e.g.*, single, k-multiple, block and block2 (*Nagata & Kobayashi, 2013*)), where an E-set is a set of AB-cycles.

4. Copy parent $S_A$ to an intermediate solution $o$. Then, remove the edges of $E_A$ in the E-set from $o$ and add those of $E_B$ in the E-set to $o$. This leads to an intermediate solution $o$ with one or more subtours.

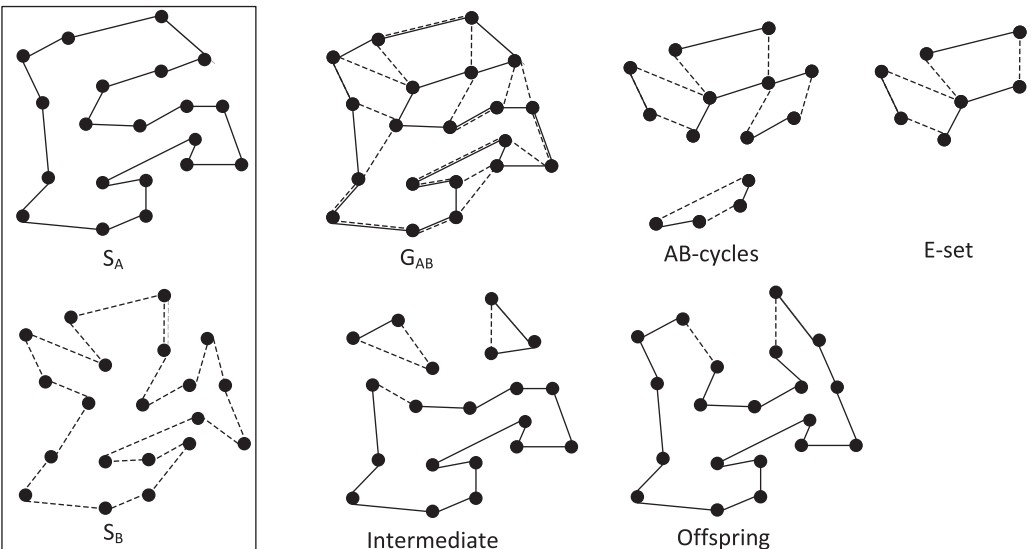

**Figure 2 Illustrative example of the EAX crossover operator.**

5. Connect all the subtours in *o* with new short edges to generate a complete tour (a feasible offspring solution) by using a greedy heuristic.

Note that different versions of EAX can be developed by using different selection strategies of AB-cycles for constructing E-sets. The GA-EAX algorithm employs the single and block2 strategies to generates offspring solutions from parent solutions. To maintain a healthy population diversity, GA-EAX also uses an edge entropy measure to select the solution to be used to replace a parent in the population.

Other studies (*e.g., Hains, Whitley & Howe, 2012*) also indicated the usefulness of edge-assembly-like crossovers for solving clustered instances of the TSP. As shown in the next section, the EAX-based genetic algorithm performs remarkably well on all the clustered instances transformed from the CTSP.

## COMPUTATIONAL EXPERIMENTS

In this section, we evaluate the capacity of the TSP solvers presented in "Solution Methods for the TSP" to solve the CTSP *via* its transformation to the TSP. For this purpose, we examine their qualitative performances and run time efficiencies on various benchmark instances and make comparisons with the best dedicated CTSP algorithms in the literature.

### Benchmark instances

Our computational assessments are based on three sets of 73 benchmark instances with 101 to 24,978 vertices. Sets 1 and 2 include 20 medium instances ($101 \leq |V| \leq 1,000$) and 15 large instances ($1,173 \leq |V| \leq 2,000$), which are classical and widely used in the CTSP literature (*e.g., Mestria, Ochi & de Lima Martins, 2013; Mestria, 2016; Mestria, 2018*). Set 3 includes 38 large GTSP instances ($1,000 \leq |V| \leq 24,978$) from *Helsgaun (2014)*.

**Sets 1 and 2 (35 instances)**: These instances belong to the following six types: (1) instances taken from the TSPLIB (*Reinelt, 1991*) where the clusters are generated by using a k-means clustering algorithm; (2) instances created from a selection of classic TSP instances (*Johnson & McGeoch, 2007*), where the clusters are created by grouping the vertices in geometric centers; (3) instances generated by using the Concorde interface (*Applegate, Bixby & Chvatal, 2006*); (4) instances generated using the layout proposed in *Laporte & Palekar (2002)*; (5) instances similar to type 2, but generated with different parameters; (6) instances adapted from the TSPLIB (*Reinelt, 1991*), where the rectangular floor plan is divided into several quadrilaterals and each quadrilateral corresponds to a cluster. These instances are available at http://www2.ic.uff.br/~labic/conteudo/instance/.

**Set 3 (38 instances)**: These large instances have 1,000 to 24,978 vertices and come from GTSPLIB for the generalized traveling salesman problem (GTSP). They were generated from TSP instances by using *Fischetti, Salazar González & Toth (1997)* clustering algorithm and tested in *Helsgaun (2014)* by considering them as CTSP instances. These instances are available at http://www.ruc.dk/~keld/research/CLKH. In *Helsgaun (2014)*, six very large instances with 31,623 to 85,900 vertices were also tested. We ignore these instances, because they are too large for the exact Concorde solver and the GA-EAX solver stops abnormally when solving these instances.

## TSP solvers and experimental protocol

For our study, we employed three representative TSP solvers presented in "Solution Methods for the TSP", which are among the most powerful methods for the TSP in the literature.

- Exact Concorde TSP solver (http://www.math.uwaterloo.ca/tsp/concorde/index.html): We used version Concorde-03.12.19 and ran the solver with its default parameter setting with a cutoff time of 24 CPU hours per instance.
- Inexact CLKH solver (http://www.ruc.dk/~keld/research/CLKH): We used the version CLKH-1.0 which is based on the latest version 2.0.9 (http://akira.ruc.dk/~keld/research/LKH/) of LKH-2. The default parameter setting given in *Helsgaun (2014)* was adopted to run CLKH. Notice that to reduce its run time, the maximum number of trials (iterations) is set to 1,000 in CLKH, while this number is set to $n$ (instance size) by default in LKH-2.
- Inexact GA-EAX TSP solver (https://github.com/sugia/GA-for-TSP): We used GA-EAX with its default parameter setting given in *Nagata & Kobayashi (2013)*: $p = 300$, $r = 30$ and GA-EAX terminates if the difference between the average tour length and the shortest tour length in the population is less than 0.001. Following *Kerschke et al. (2018)* and *Kotthoff et al. (2015)*, we reset the random seed for GA-EAX for each run (which was set to a fixed value in the official implementation).

The experiments were carried out on a computer running Linux operating system with an Intel E5-2670 processor (2.8 GHz and 4G RAM). Given the stochastic nature of CLKH

**Table 1** Computational results of the TSP solvers Concorde, CLKH and GA-EAX on medium CTSP instances (Set 1).

| Instance | $|V|$ | $m$ | Concorde | | CLKH | | | GA-EAX | | |
|---|---|---|---|---|---|---|---|---|---|---|
| | | | Opt. | $t(s)$ | $Gap_{best}$ | $Gap_{avg}$ | $t(s)$ | $Gap_{best}$ | $Gap_{avg}$ | $t(s)$ |
| i-50-gil262 | 262 | 50 | 135,431 | 1.9 | =(10) | 0.0000 | 1.3 | =(10) | 0.0000 | 1.7 |
| 10-lin318 | 318 | 10 | 529,584 | 2.2 | =(10) | 0.0000 | 19.5 | =(10) | 0.0000 | 1.8 |
| 10-pcb442 | 442 | 10 | 537,419 | 20.7 | =(10) | 0.0000 | 46.9 | =(10) | 0.0000 | 6.3 |
| C1k.0 | 1,000 | 10 | 132,521,027 | 21.9 | =(9) | 0.0001 | 128.6 | =(10) | 0.0000 | 16.3 |
| C1k.1 | 1,000 | 10 | 129,128,125 | 22.3 | =(10) | 0.0000 | 70.6 | =(10) | 0.0000 | 14.3 |
| C1k.2 | 1,000 | 10 | 142,784,000 | 69.9 | 0.0009 | 0.0009 | 244.6 | =(9) | 0.0001 | 17.2 |
| 300-6 | 300 | 6 | 8,934 | 4.4 | =(10) | 0.0000 | 30.2 | =(10) | 0.0000 | 3.5 |
| 400-6 | 400 | 6 | 9,045 | 6.7 | =(10) | 0.0000 | 26.7 | =(10) | 0.0000 | 4.4 |
| 700-20 | 700 | 20 | 41,425 | 29.9 | =(10) | 0.0000 | 200.0 | =(10) | 0.0000 | 10.2 |
| 200-4-h | 200 | 4 | 62,777 | 0.6 | =(10) | 0.0000 | 5.4 | =(10) | 0.0000 | 0.9 |
| 200-4-x1 | 200 | 4 | 60,574 | 1.1 | =(10) | 0.0000 | 6.5 | =(10) | 0.0000 | 0.9 |
| 600-8-z | 600 | 8 | 128,891 | 9.9 | =(10) | 0.0000 | 48.2 | =(10) | 0.0000 | 5.3 |
| 600-8-x2 | 600 | 8 | 128,891 | 4.8 | =(10) | 0.0000 | 48.2 | =(10) | 0.0000 | 5.3 |
| 300-5-108 | 300 | 5 | 67,760 | 1.2 | =(10) | 0.0000 | 8.5 | =(10) | 0.0000 | 2.0 |
| 300-20-111 | 300 | 20 | 309,739 | 1.8 | =(10) | 0.0000 | 6.0 | =(10) | 0.0000 | 2.0 |
| 500-15-306 | 500 | 15 | 194,818 | 2.6 | =(10) | 0.0000 | 37.1 | =(10) | 0.0000 | 5.2 |
| 500-25-308 | 500 | 25 | 365,447 | 4.4 | =(10) | 0.0000 | 10.1 | =(10) | 0.0000 | 5.4 |
| 25-eil101 | 101 | 25 | 23,671 | 0.5 | =(10) | 0.0000 | 0.4 | =(10) | 0.0000 | 0.8 |
| 42-a280 | 280 | 42 | 129,645 | 2.3 | =(10) | 0.0000 | 2.4 | =(10) | 0.0000 | 1.7 |
| 144-rat783 | 783 | 144 | 914,228 | 70.2 | =(10) | 0.0000 | 14.6 | =(10) | 0.0000 | 9.4 |
| Avg. | | | | 14.0 | | 0.0001 | 47.8 | | 0.0000 | 5.7 |

and GA-EAX, we ran each algorithm 10 times for each instances while the deterministic Concorde TSP solver was run one time to solve each instance.

## Computational results and comparison of popular TSP solvers

Our computational studies aim to answer the following questions: How do state-of-the-art *exact* TSP solvers perform on clustered instances converted from the CTSP? How do state-of-the-art *inexact* (heuristic) TSP solvers perform on clustered instances converted from the CTSP?

The results of the three TSP solvers (Concorde, CLKH, GA-EAX) on the 20 medium and 15 large CTSP benchmark instances are summarized in Table 1 (Set 1) and Table 2 (Set 2). Columns 1 to 3 show the basic information of each instance: the instance name (Instance), the number of vertices ($|V|$) and the number of clusters ($m$). Column 4 gives the optimal objective value reported by the exact Concorde TSP solver, followed by the required run time in seconds. For both the CLKH and GA-EAX solvers, we show the best ($Gap_{best}$) and average ($Gap_{avg}$) results over 10 independent runs in the form of the percentage gap to the optimal solution, as well as the average run time in seconds. If the best solution over 10 independent runs equals the optimal solution obtained with the

**Table 2 Computational results of the TSP solvers Concorde, CLKH and GA-EAX on large CTSP instances (Set 2).**

| Instance | $|V|$ | $m$ | Concorde | | CLKH | | | GA-EAX | | |
|---|---|---|---|---|---|---|---|---|---|---|
| | | | Opt. | $t(s)$ | $Gap_{best}$ | $Gap_{avg}$ | $t(s)$ | $Gap_{best}$ | $Gap_{avg}$ | $t(s)$ |
| 49-pcb1173 | 1,173 | 49 | 61,600 | 5,638.3 | 0.6250 | 1.0519 | 1,065.8 | =(4) | 0.0326 | 35.0 |
| 100-pcb1173 | 1,173 | 100 | 63,382 | 588.3 | =(7) | 0.0066 | 63.2 | =(8) | 0.0013 | 32.5 |
| 144-pcb1173 | 1,173 | 144 | 62,142 | 38.4 | =(10) | 0.0000 | 25.8 | =(10) | 0.0000 | 18.6 |
| 10-nrw1379 | 1,379 | 10 | 58,783 | 562.9 | =(10) | 0.0000 | 174.9 | =(6) | 0.0070 | 26.8 |
| 12-nrw1379 | 1,379 | 12 | 59,129 | 58.5 | =(10) | 0.0000 | 39.7 | =(9) | 0.0007 | 27.6 |
| 1500-10-503 | 1,500 | 10 | 11,116 | 65.5 | =(5) | 0.0225 | 603.6 | =(10) | 0.0000 | 28.4 |
| 1500-20-504 | 1,500 | 20 | 15,698 | 40.7 | =(10) | 0.0000 | 167.9 | =(5) | 0.0172 | 34.5 |
| 1500-50-505 | 1,500 | 50 | 22,900 | 67.0 | =(7) | 0.0476 | 178.8 | =(5) | 0.0044 | 35.1 |
| 1500-100-506 | 1,500 | 100 | 29,799 | 108.7 | =(6) | 0.0228 | 58.3 | =(8) | 0.0020 | 39.5 |
| 1500-150-507 | 1,500 | 150 | 34,068 | 114.7 | =(10) | 0.0000 | 44.4 | =(10) | 0.0000 | 32.3 |
| 2000-10-a | 2,000 | 10 | 105,360 | 7214.3 | 0.0038 | 0.0155 | 401.9 | 0.0826 | 0.1167 | 45.3 |
| 2000-10-h | 2,000 | 10 | 33,708 | 812.7 | =(9) | 0.0006 | 229.9 | =(10) | 0.0000 | 35.6 |
| 2000-10-z | 2,000 | 10 | 33,509 | 200.9 | =(10) | 0.0000 | 160.1 | =(9) | 0.0003 | 37.3 |
| 2000-10-x1 | 2,000 | 10 | 33,792 | 1,325.4 | =(4) | 0.0213 | 485.3 | =(6) | 0.0136 | 35.6 |
| 2000-10-x2 | 2,000 | 10 | 33,509 | 170.9 | =(10) | 0.0000 | 160.1 | =(10) | 0.0000 | 39.6 |
| Avg. | | | | 1,133.8 | | 0.0793 | 257.3 | | 0.0131 | 33.6 |

exact Concorde TSP solver, the corresponding cell in column $Gap_{best}$ shows '=' along with the number of runs that succeeded in finding the optimal solution. Finally, row 'Avg.' provides the average run time in seconds for each approach, and the average gap between the average objective values obtained with CLKH/GA-EAX and the optimal values obtained with the Concorde TSP solver.

From Tables 1 to 2, we can make the following observations:

First, the exact Concorde TSP solver performs very well on these 35 instances and is able to solve all of them exactly. Specifically, the 20 medium instances can be solved easily in a short run time (an average of about 14 s). The 15 large instances are more difficult and the run time needed to solve these instances increases considerably (an average of 1,133.8 s, reaching 7,214.3 s for the most difficult instance).

Second, the CLKH solver performs globally very well on these 35 instances. For the 20 medium instances, CLKH attains all the optimal solutions but one with an average run time of 47.8. For the 15 large instances, CLKH reaches the optimal solutions for 13 instances with an average run time of 257.3 s.

Third, the GA-EAX solver performs remarkably well by attaining the optimal values for all 35 instances but one. For the 20 medium instances, GA-EAX consistently hits the optimal solutions for each of its 10 run (except for one instance for which it has a hit of 9 out of 10). The average run time is only 5.7 s for the medium instances and 33.6 s for the large instances. Compared to Concorde and CLKH, GA-EAX is thus extremely time efficient. Moreover, contrary to the Concorde and CLKH solvers, the computation time

required by GA-EAX remains very stable across the instances of the same set, indicating a high robustness and scalability of this solver.

Table 3 presents the results of the three TSP solvers on the 38 large GTSP instances of Set 3. Notice that the Concorde solver failed to exactly solve 17 instances in 24 h, the corresponding cell (in parentheses) in column 'Optimum' indicates the best tour length (best upper bound) found by CLKH and GA-EAX. In this case, the percentage gaps ($Gap_{best}$ and $Gap_{avg}$) are calculated by using the best bound, and column $Gap_{best}$ shows '=' the number of runs for an algorithm to find the best bound.

From Table 3, we can make the following observations. First, Concorde manages to optimally solve 21 large GTSP instances with up to 3,162 vertices with a run time ranging from 17.4 s to 45,008.4 s while its solution time is not completely consistent with the size of the problem instances. For the 21 instances that can be solved exactly by Concorde, CLKH attains 15 best upper bounds, while GA-EAX reaches all best upper bounds in less computing time. Second, for most of the instances with $|V| < 10,000$, compared with CLKH, GA-EAX has a better performance both in terms of solution quality and computation time. For the instances with $10,000 \leq |V| \leq 24,978$, the solution quality of GA-EAX is better than that of CLKH in most cases, while requiring more computation time.

To sum up, the exact Concorde solver is very efficient for the instances with up to 1,000 vertices (order of seconds) and can even find optimal solutions for instances with up to some 3,000 vertices at a price of more run time (order of minutes to hours). For larger instances, both inexact solvers (CLKH and GA-EAX) are reliable alternatives to find optimal or sub-optimal solutions with some advantages for GA-EAX. These heuristic solvers also perform very well on smaller instances.

To deepen our computational study, we call upon to the performance profile, an analytic tool for evaluating the performances of multiple compared optimization algorithms (*Dolan & Moré, 2002*). The performance profile uses a cumulative distribution function for a performance metric, such as run time, objective function values, number of iterations, and so on. Precisely, let $S$ be a set of algorithms and $P$ be a set of problem instances. For a given performance metric $f_{s,p}$ (that is the performance of algorithm $s \in S$ solving instance $p \in P$), the performance ratio is defined by $r_{s,p} = \dfrac{f_{s,p}}{min\{f_{a,p} : a \in S\}}$. Then, for each algorithm $s \in S$, the performance function is given by $\rho_s(\tau) = \dfrac{|\{p \in P : r_{s,p} \leq \tau\}|}{|P|}$.

Thus, the value of $\rho_s(1)$ corresponds to the fraction of problem instances that algorithm $s$ can achieve many times the performance of the best algorithm (meaning the probability that the algorithm $s$ will win over the rest of the compared algorithms). For a large value $\tau$, the value of $\rho_s(\tau)$ indicates a high robustness of algorithm $s$.

To make a fair and meaningful comparison with this tool, we focus on the two inexact solvers CLKH and GA-EAX and run each solver 10 times on each of the 73 instances. We use the software 'perprof-py' (*Siqueira, da Silva & Santos, 2016*) to draw the

**Table 3 Computational results of the TSP solvers Concorde, CLKH and GA-EAX on large GTSP instances (Set 3).**

| Instance | \|V\| | m | Optimum | Concorde's run-time | CLKH $Gap_{best}$ | $Gap_{avg}$ | t(s) | GA-EAX $Gap_{best}$ | $Gap_{avg}$ | t(s) |
|---|---|---|---|---|---|---|---|---|---|---|
| 10C1k.0 | 1,000 | 10 | 12,139,627 | 23.5 | =(9) | 0.0016 | 194.5 | =(10) | 0.0000 | 16.1 |
| 200C1k.0 | 1,000 | 200 | 11,929,315 | 17.4 | =(10) | 0.0000 | 64.7 | =(10) | 0.0000 | 15.6 |
| 200E1k.0 | 1,000 | 200 | 24,468,822 | 66.2 | =(8) | 0.0008 | 27.7 | =(10) | 0.0000 | 15.1 |
| 49usa1097 | 1,097 | 49 | 77,583,052 | 51.1 | =(7) | 0.0069 | 128.6 | =(10) | 0.0000 | 23.7 |
| 235pcb1173 | 1,173 | 235 | 59,796 | 65.5 | =(9) | 0.0151 | 36.4 | =(10) | 0.0000 | 16.4 |
| 259d1291 | 1,291 | 259 | 55,962 | 8,402.5 | 0.0286 | 0.0484 | 51.7 | =(7) | 0.0064 | 17.3 |
| 261rl1304 | 1,304 | 261 | 261,132 | 19.2 | =(10) | 0.0000 | 18.7 | =(10) | 0.0000 | 7.5 |
| 265rl1323 | 1,323 | 265 | 280,004 | 3,361.3 | 0.0114 | 0.0381 | 18.9 | =(8) | 0.0019 | 10.2 |
| 276nrw1379 | 1,379 | 276 | 60,473 | 234.4 | =(3) | 0.0223 | 30.8 | =(10) | 0.0000 | 30.7 |
| 280fl1400 | 1,400 | 280 | 20,229 | 6,108.5 | =(3) | 0.0900 | 504.7 | =(8) | 0.0178 | 21.5 |
| 287u1432 | 1,432 | 287 | 162,151 | 23,029.9 | =(8) | 0.0136 | 111.6 | =(10) | 0.0000 | 26.8 |
| 316fl1577 | 1,577 | 316 | 23,023 | 1,179.6 | =(10) | 0.0000 | 183.0 | =(2) | 0.2332 | 17.2 |
| 331d1655 | 1,655 | 331 | 65,871 | 142.9 | =(3) | 0.0797 | 51.8 | =(7) | 0.0029 | 24.6 |
| 350vm1748 | 1,748 | 350 | 348,244 | 230.9 | =(2) | 0.0371 | 88.2 | =(10) | 0.0000 | 25.7 |
| 364u1817 | 1,817 | 364 | 61,879 | 5,675.7 | =(1) | 0.0739 | 77.7 | =(6) | 0.0050 | 31.8 |
| 378rl1889 | 1,889 | 378 | 323,040 | 461.5 | =(1) | 0.1197 | 29.0 | =(10) | 0.0000 | 18.1 |
| 421d2103 | 2,103 | 421 | (91,637) | – | =(2) | 0.0598 | 112.7 | =(10) | 0.0000 | 32.8 |
| 431u2152 | 2,152 | 431 | (69,876) | – | =(2) | 0.0215 | 98.5 | =(10) | 0.0000 | 37.0 |
| 464u2319 | 2,319 | 464 | (246,707) | – | =(10) | 0.0000 | 703.2 | =(3) | 0.0167 | 84.9 |
| 479pr2392 | 2,392 | 479 | 397,707 | 1,267.5 | =(4) | 0.0223 | 102.1 | =(10) | 0.0000 | 38.0 |
| 608pcb3038 | 3,038 | 608 | 146,351 | 45,008.4 | 0.0014 | 0.0256 | 115.5 | =(4) | 0.0018 | 83.2 |
| 31C3k.0 | 3,162 | 31 | 20,058,457 | 912.6 | 0.0144 | 0.0637 | 249.2 | =(5) | 0.0211 | 111.6 |
| 633C3k.0 | 3,162 | 633 | 20,158,425 | 1,650.4 | 0.0207 | 0.0869 | 163.5 | =(8) | 0.0011 | 98.0 |
| 633E3k.0 | 3,162 | 633 | 42,697,510 | 5,239.0 | 0.0036 | 0.0226 | 105.7 | =(3) | 0.0052 | 115.0 |
| 759fl3795 | 3,795 | 759 | (29,582) | – | =(9) | 0.0068 | 464.0 | 0.2637 | 0.3729 | 53.9 |
| 893fnl4461 | 4,461 | 893 | (193,834) | – | =(2) | 0.0163 | 139.0 | =(8) | 0.0004 | 236.6 |
| 1183rl5915 | 5,915 | 1,183 | (599,096) | – | 0.0212 | 0.1666 | 204.3 | =(9) | 0.0006 | 146.6 |
| 1187rl5934 | 5,934 | 1,187 | (588,074) | – | 0.0126 | 0.1256 | 251.6 | =(5) | 0.0033 | 156.1 |
| 1480pla7397 | 7,397 | 1,480 | (23,926,551) | – | 0.0035 | 0.0213 | 1104.7 | =(2) | 0.0078 | 388.2 |
| 100C10k.0 | 10,000 | 100 | (36,352,580) | – | =(1) | 0.6815 | 1877.4 | 0.0525 | 0.4872 | 2318.7 |
| 2000C10k.0 | 10,000 | 2,000 | (34,574,383) | – | 0.0369 | 0.2590 | 730.6 | =(1) | 0.0139 | 992.9 |
| 2000E10k.0 | 10,000 | 2,000 | (75,506,665) | – | 0.0112 | 0.0281 | 635.8 | =(1) | 0.0013 | 1,320.0 |
| 2370rl11849 | 11,849 | 2,370 | (977,472) | – | 0.0081 | 0.0477 | 757.1 | =(1) | 0.0028 | 1,051.9 |
| 2702usa13509 | 13,509 | 2,702 | (20,836,160) | – | 0.0118 | 0.0185 | 1028.6 | =(1) | 0.0012 | 2,154.0 |
| 2811brd14051 | 14,051 | 2,811 | (496,827) | – | 0.0125 | 0.0213 | 944.7 | =(1) | 0.0024 | 2,454.9 |
| 3023d15112 | 15,112 | 3,023 | (1,658,091) | – | 0.0220 | 0.0296 | 1193.3 | =(1) | 0.0019 | 3,864.0 |
| 3703d18512 | 18,512 | 3,703 | (683,839) | – | 0.0209 | 0.0328 | 1561.9 | =(1) | 0.0019 | 4,306.8 |
| 4996sw24978 | 24,978 | 4,996 | (893,042) | – | 0.0237 | 0.0369 | 2076.3 | =(1) | 0.0008 | 5,706.2 |
| Avg. | | | | | | 0.0616 | 427.3 | | 0.0319 | 686.0 |

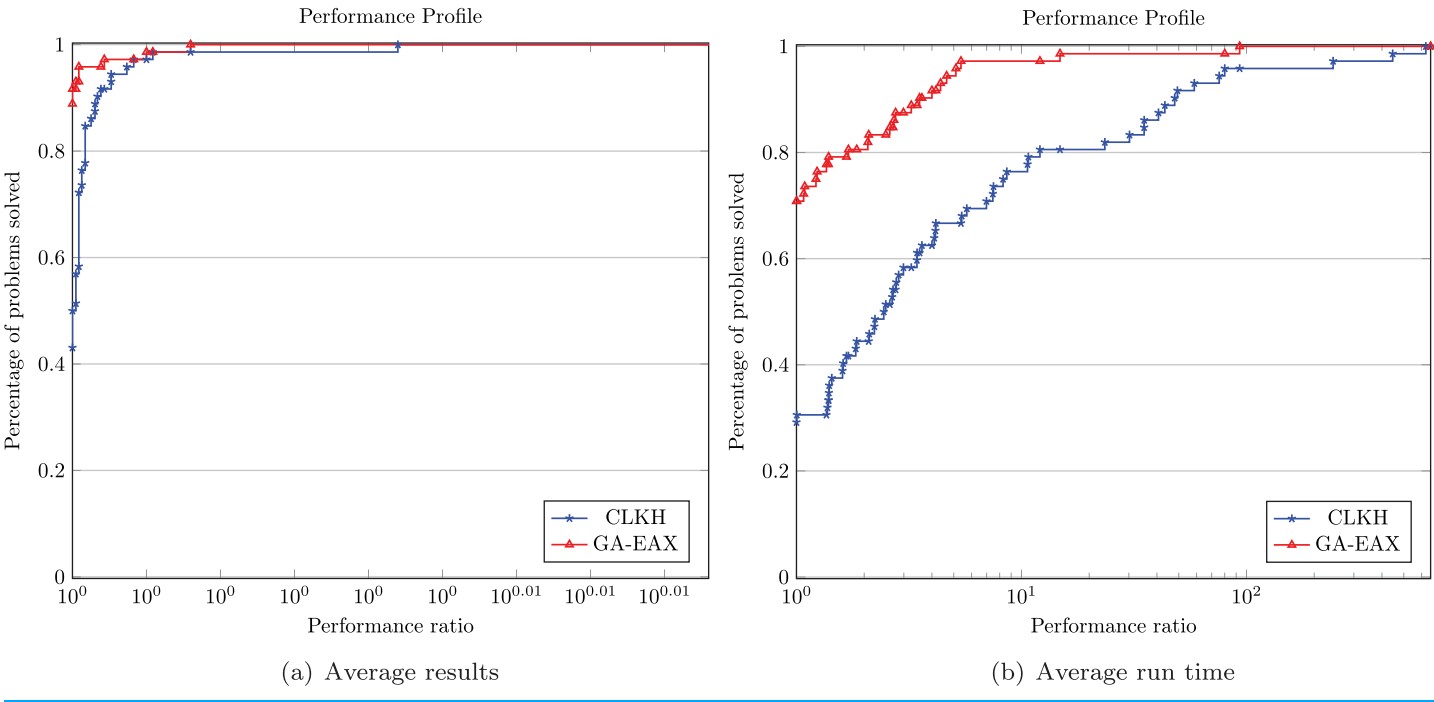

**Figure 3 Performance profiles comparing solution quality and computing time.**

performance profiles (see Fig. 3) where the quality of the solution is measured by the average objective value and average run time. These performance profiles tend to show an advantage of GA-EAX over CLKH for solving these clustered instances with up to 24,978 vertices.

## TSP solvers v.s. state-of-the-art CTSP heuristics

In "Computational Results and Comparison of Popular TSP Solvers", we observed that the exact Concord TSP solver and the inexact CLKH and GA-EAX TSP solvers are powerful tools for solving clustered TSP instances converted from the CTSP. We now answer the following question: Do these general TSP solvers compete well with state-of-the-art CTSP heuristics specially designed for the problem?

For this purpose, we adopt GA-EAX as our representative TSP solver and compare it with three best performing CTSP heuristics in the literature: VNRDGILS (*Mestria, 2018*), HHGILS (*Mestria, 2016*), and GPR1R2 (*Mestria, Ochi & de Lima Martins, 2013*). Indeed, according to the experimental studies reported in *Mestria, Ochi & de Lima Martins (2013)* and *Mestria (2016, 2018)*, these three heuristics perform the best among the recent CTSP heuristics available in the literature (see Table 4). This study is based on the 35 medium and large instances of Sets 1 and 2 (no results for the three CTSP heuristics are available on the large GTSP instances of Set 3).

Table 5 provides the comparative results of the GA-EAX TSP solver along with the results reported by the three CTSP algorithms on the medium and large instances. For each instance and algorithm, columns '$f_{best}$', '$f_{avg}$' and '$t(s)$' show respectively the best objective value over 10 independent runs, the average objective value and the average run time

**Table 4 List of the reference algorithms for the CTSP.**

| Algorithm name | Reference | Search strategy |
|---|---|---|
| VNRDGILS | *Mestria (2018)* | A hybrid heuristic based on GRASP, ILS and VNRD |
| HHGILS | *Mestria (2016)* | A hybrid heuristic based on GRASP, ILS and VND |
| GPR1R2 | *Mestria, Ochi & de Lima Martins (2013)* | A GRASP with Path Relinking PR1 and PR2 |
| GPR1 | *Mestria, Ochi & de Lima Martins (2013)* | A GRASP with Path Relinking PR1 |
| GPR2 | *Mestria, Ochi & de Lima Martins (2013)* | A GRASP with Path Relinking PR2 |
| GPR3 | *Mestria, Ochi & de Lima Martins (2013)* | A GRASP with Path Relinking PR3 |
| GPR4 | *Mestria, Ochi & de Lima Martins (2013)* | A GRASP with Path Relinking PR4 |
| GRASP | *Mestria, Ochi & de Lima Martins (2013)* | A traditional GRASP heuristic |
| TLGA | *Ding, Cheng & He (2007)* | A two-level genetic algorithm |

in seconds. Furthermore, the row 'Avg.' shows the average performances for each compared algorithm, including the average percentage gap of the best/average result to the optimal result obtained with the Concorde TSP solver and the average run time in seconds. To determine whether there exists a statistically significant difference in performance between the GA-EAX TSP solver and each CTSP algorithm in terms of best and average results, the $p$-values from the Wilcoxon signed-rank tests are given in the last row of Table 5. Entries with "-" mean that the corresponding results are not available in the literature. The best objective values obtained by the compared algorithms are indicated in bold if they attain the optimal solution. Notice that the results of the CTSP algorithms (VNRDGILS, HHGILS and GPR1R2) correspond to 10 executions per instance on a computer with 2.83 GHz Intel Core 2 CPU and 8 GB RAM and the time limit per run was set to 720 s for medium instances and 1,080 s for large instances.

Table 6 summarizes the statistical results for each compared algorithm on the two sets of medium and large instances. The first row indicates the number of optimal solutions found by each approach. The average percentage gap of the best/average result from the optimal result is provided in row 'Average $Gap_{best}$/$Gap_{avg}$'. Finally, row 'Average time (s)' provides the average run time in seconds for each algorithm.

From Tables 5 and 6, we observe that the GA-EAX solver significantly outperforms the three CTSP algorithms on the medium and large instances in terms of both the best and the average results. For the large instance set, the improvement gaps between the results of GA-EAX and those of the CTSP methods are very high, ranging from 10.39% to 15.49%. Furthermore, in terms of the average run time, GA-EAX is about 30 to 130 times faster than the CTSP algorithms. The above results thus indicate that the GA-EAX TSP solver has a strong dominance over current best performing CTSP approaches in the literature. In addition, the small $p$-values ($<0.05$) from the Wilcoxon signed-rank tests further confirm the statistically significant difference of the compared results.

To have a finer analysis of the compared algorithms, Fig. 4 provides boxplot graphs to compare the distribution and range of the average results for each compared algorithm, except GPR1R2 for the medium instances since its results on several medium instances are not available. In this figure, the average objective value $f_{avg}$ of a given algorithm is

**Table 5 Comparative results between the GA-EAX TSP solver and three CTSP algorithms on medium and large CTSP instances.** The best objective values obtained by the compared algorithms are indicated in bold if they attain the optimal solution.

| | | | GA-EAX | | | VNRDGILS | | | HHGILS | | | GPR1R2 | | |
|---|---|---|---|---|---|---|---|---|---|---|---|---|---|---|
| Instance | $\|V\|$ | $m$ | $f_{best}$ | $f_{avg}$ | $t(s)$ | $f_{best}$ | $f_{avg}$ | $t(s)$ | $f_{best}$ | $f_{avg}$ | $t(s)$ | $f_{best}$ | $f_{avg}$ | $t(s)$ |
| i-50-gil262 | 262 | 50 | **135,431** | 135,431.0 | 1.7 | 135,483 | 135,510.2 | 720.0 | 135,510 | 135,578 | 720.0 | – | – | – |
| 10-lin318 | 318 | 10 | **529,584** | 529,584.0 | 1.8 | 530,604 | 530,871.4 | 720.0 | 530,283 | 530,817.9 | 720.0 | 530,443 | 532,697.9 | 720.0 |
| 10-pcb442 | 442 | 10 | **537,419** | 537,419.0 | 6.3 | 538,309 | 538,903.4 | 720.0 | 538,958 | 539,988.3 | 720.0 | 540,043 | 543,104.2 | 720.0 |
| C1k.0 | 1,000 | 10 | **132,521,027** | 132,521,027.0 | 16.3 | 133,260,549 | 133,490,775.9 | 720.0 | 133,287,594 | 133,776,274.1 | 720.0 | 133,490,776 | 133,708,187.6 | 720.0 |
| C1k.1 | 1,000 | 10 | **129,128,125** | 129,128,125.0 | 14.3 | 129,877,874 | 130,035,540.2 | 720.0 | 129,825,403 | 130,206,778.3 | 720.0 | 130,193,590 | 130,391,693.5 | 720.0 |
| C1k.2 | 1,000 | 10 | **142,784,000** | 142,784,188.4 | 17.2 | 143,321,630 | 143,481,489.6 | 720.0 | 143,278,093 | 143,525,149.6 | 720.0 | – | – | – |
| 300-6 | 300 | 6 | **8,934** | 8,934.0 | 3.5 | 8,935 | 8,941.1 | 720.0 | **8,934** | 8,942.9 | 720.0 | 8959 | 8,985.3 | 720.0 |
| 400-6 | 400 | 6 | **9,045** | 9,045.0 | 4.4 | 9,053 | 9,062.3 | 720.0 | 9,051 | 9,063.2 | 720.0 | – | – | – |
| 700-20 | 700 | 20 | **41,425** | 41,425.0 | 10.2 | 41,456 | 41,489.7 | 720.0 | 41,452 | 41,485.6 | 720.0 | 41,540 | 41,573.3 | 720.0 |
| 200-4-h | 200 | 4 | **62,777** | 62,777.0 | 0.9 | 62,867 | 63,058.3 | 720.0 | 62,804 | 63,058.3 | 720.0 | 62,994 | 63,710.2 | 720.0 |
| 200-4-x1 | 200 | 4 | **60,574** | 60,574.0 | 0.9 | 60,637 | 60,796.2 | 720.0 | 60,931 | 61,378.5 | 720.0 | – | – | – |
| 600-8-z | 600 | 8 | **128,891** | 128,891.0 | 5.3 | 129,468 | 129,862.7 | 720.0 | 129,416 | 129,928.6 | 720.0 | 130,459 | 131,235.1 | 720.0 |
| 600-8-x2 | 600 | 8 | **128,891** | 128,891.0 | 5.3 | 129,246 | 129,533.9 | 720.0 | 129,246 | 129,691.5 | 720.0 | – | – | – |
| 300-5-108 | 300 | 5 | **67,760** | 67,760.0 | 2.0 | 67,766 | 67,868.7 | 720.0 | 67,814 | 67,930.5 | 720.0 | – | – | – |
| 300-20-111 | 300 | 20 | **309,739** | 309,739.0 | 2.0 | 310,146 | 310,270.9 | 720.0 | 310,209 | 310,427 | 720.0 | 309,928 | 310,551.9 | 720.0 |
| 500-15-306 | 500 | 15 | **194,818** | 194,818.0 | 5.2 | 194,946 | 195,201.5 | 720.0 | 195,202 | 195,438.1 | 720.0 | – | – | – |
| 500-25-308 | 500 | 25 | **365,447** | 365,447.0 | 5.4 | 365,717 | 365,937.8 | 720.0 | 365,828 | 366,085 | 720.0 | 366,232 | 366,785.7 | 720.0 |
| 25-eil101 | 101 | 25 | **23,671** | 23,671.0 | 0.8 | 23,673 | 23,685.2 | 720.0 | 23,678 | 23,690 | 720.0 | 23,676 | 23,711.3 | 720.0 |
| 42-a280 | 280 | 42 | **129,645** | 129,645.0 | 1.7 | 129,729 | 129,755.2 | 720.0 | 129,716 | 129,833.2 | 720.0 | – | – | – |
| 144-rat783 | 783 | 144 | **914,228** | 914,228.0 | 9.4 | 915,088 | 915,179.8 | 720.0 | 915,180 | 915,363.2 | 720.0 | 915,547 | 915,913.7 | 720.0 |
| 49-pcb1173 | 1,173 | 49 | **61,600** | 61,620.1 | 35.0 | 65,750 | 66,487.7 | 1,080.0 | 67,043 | 68,260.7 | 1,080.0 | 70,651 | 73,311.9 | 1,080.0 |
| 100-pcb1173 | 1,173 | 100 | **63,382** | 63,382.8 | 32.5 | 68,708 | 69,383.2 | 1,080.0 | 68,786 | 70,640.8 | 1,080.0 | 72,512 | 74,871.7 | 1,080.0 |
| 144-pcb1173 | 1,173 | 144 | **62,142** | 62,142.0 | 18.6 | 68,414 | 68,941.4 | 1,080.0 | 66,830 | 69,084.3 | 1,080.0 | 72,889 | 74,621.6 | 1,080.0 |
| 10-nrw1379 | 1,379 | 10 | **58,783** | 58,787.1 | 26.8 | 63,951 | 64,895.9 | 1,080.0 | 63,620 | 64,643.9 | 1,080.0 | 66,747 | 68,955.8 | 1,080.0 |
| 12-nrw1379 | 1,379 | 12 | **59,129** | 59,129.4 | 27.6 | 62,893 | 63,532.3 | 1,080.0 | 63,558 | 64,741.6 | 1,080.0 | 66,444 | 69,141.2 | 1,080.0 |
| 1500-10-503 | 1,500 | 10 | **11,116** | 11,116.0 | 28.4 | 11,969 | 12,103.0 | 1,080.0 | 11,986 | 12,109.5 | 1,080.0 | 12,278 | 12,531.4 | 1,080.0 |
| 1500-20-504 | 1,500 | 20 | **15,698** | 15,700.7 | 34.5 | 16,678 | 16,867.4 | 1,080.0 | 17,107 | 17,315.7 | 1,080.0 | 17,252 | 17,589.1 | 1,080.0 |
| 1500-50-505 | 1,500 | 50 | **22,900** | 22,901.0 | 35.1 | 24,631 | 24,803.6 | 1,080.0 | 25,264 | 25,558.9 | 1,080.0 | 25,124 | 25,761.5 | 1,080.0 |
| 1500-100-506 | 1,500 | 100 | **29,799** | 29,799.6 | 39.5 | 32,474 | 32,616.8 | 1,080.0 | 32,260 | 33,760.6 | 1,080.0 | 33,110 | 33,692.7 | 1,080.0 |
| 1500-150-507 | 1,500 | 150 | **34,068** | 34,068.0 | 32.3 | 37,357 | 38,251.1 | 1,080.0 | 37,658 | 38,433.1 | 1,080.0 | 38,767 | 39,478.0 | 1,080.0 |
| 2000-10-a | 2,000 | 10 | 105,447 | 105,483.0 | 45.3 | 115,779 | 116,897.3 | 1,080.0 | 116,254 | 116,881.4 | 1,080.0 | 116,473 | 118,297.5 | 1,080.0 |
| 2000-10-h | 2,000 | 10 | **33,708** | 33,708.0 | 35.6 | 36,806 | 38,351.8 | 1,080.0 | 36,447 | 37,305.1 | 1,080.0 | 37,529 | 38,861.8 | 1,080.0 |
| 2000-10-z | 2,000 | 10 | **33,509** | 33,509.1 | 37.3 | 36,815 | 38,035.7 | 1,080.0 | 37,059 | 37,443.7 | 1,080.0 | 37,440 | 38,765.9 | 1,080.0 |
| 2000-10-x1 | 2,000 | 10 | **33,792** | 33,796.6 | 35.6 | 36,783 | 37,488.6 | 1,080.0 | 36,752 | 37,704.0 | 1,080.0 | 37,262 | 39,253.1 | 1,080.0 |
| 2000-10-x2 | 2,000 | 10 | **33,509** | 33,509.0 | 39.6 | 37,132 | 38,240.6 | 1,080.0 | 36,660 | 37,117.1 | 1,080.0 | 37,704 | 38,699.5 | 1,080.0 |
| Avg. | | | 0.00 | 0.01 | 17.7 | 3.79 | 4.62 | 874.3 | 3.94 | 4.96 | 874.3 | 6.98 | 8.94 | 920.0 |
| p-value | | | | | | 2.477e−7 | 2.477e−7 | | 3.651e−7 | 2.477e−7 | | | | |

normalized according to the relation $y = 100 * (f_{avg} - f_{opt})/f_{opt}$, where $f_{opt}$ is the optimal value. The plots in Fig. 4 show clear differences in the distributions of the average results between GA-EAX and each compared CTSP heuristic, which further confirms the efficiency of the GA-EAX TSP solver with respect to these dedicated CTSP heuristics.

**Table 6** Statistical results for the GA-EAX TSP solver and three state-of-the-art CTSP algorithms on Set 1 (medium instances) and Set 2 (large instances). Dominating values are indicated in bold.

| | | GA-EAX | VNRDGILS | HHGILS | GPR1R2 |
|---|---|---|---|---|---|
| Set 1 | Optimal solutions | **20/20** | 0/20 | 1/20 | 0/20 |
| | Average $Gap_{best}/Gap_{avg}$ (%) | **0.00/0.00** | 0.18/0.30 | 0.21/0.40 | 0.39/0.73 |
| | Average time (s) | **5.7** | 720.0 | 720.0 | 720.0 |
| Set 2 | Optimal solutions | **14/15** | 0/15 | 0/15 | 0/15 |
| | Average $Gap_{best}/Gap_{avg}$ (%) | **0.00/0.01** | 8.61/10.39 | 8.92/11.04 | 12.25/15.51 |
| | Average time (s) | **33.6** | 1,080.0 | 1,080.0 | 1,080.0 |

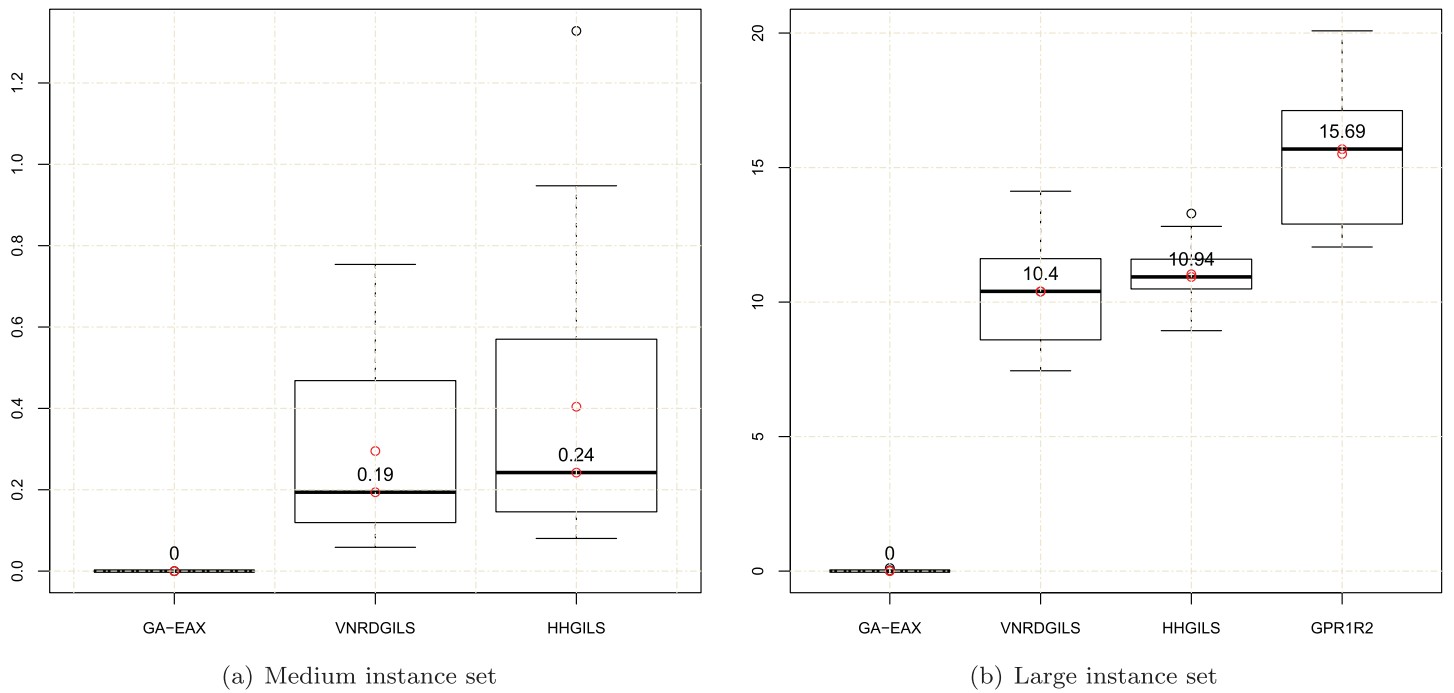

(a) Medium instance set           (b) Large instance set

**Figure 4** Boxplots of the normalized average objective values for the (A) medium instance set and (B) large instance set.

Finally, considering the results of the Concorde solver and the CLKH solver reported in "Computational Results and Comparison of Popular TSP Solvers", we conclude that these TSP solvers also dominate the current best CTSP algorithms in the literature.

## DISCUSSION

We now provide additional explanations regarding the behaviors of the three TSP solvers. First, given the NP-hard nature of the CTSP and the exponential time complexity of the exact Concorde solver, it is expected that the exact Concorde solver reaches its limit when the instance to be solved reaches some size (about 3,000 vertices for the studies instances). Indeed, when the search space becomes extremely large, the exact Branch-and-

Bound search even equipped with the best problem specific cutting plane methods cannot effectively enumerate all candidate solutions. In fact, such a behavior has already been observed in previous studies on Concorde applied to classical TSP instances (*Applegate et al., 2006*; *Hoos & Stützle, 2014*). Second, regarding the two heuristic solvers CLKH and GA-EAX, the CLKH solver exhibits a worse performance compared to GA-EAX. As discussed in "Lin-Kernighan Based Heuristic Solver", the underlying LK heuristic stumbles on clustered instances because relatively large intercluster edges serve as bait edges. With the presence of these bait edges, the LK heuristic may be tricked into long and often fruitless search trajectories. Third, the GA-EAX solver performs its search mainly with its edge assembly crossover, which inherits the edges of the parents to construct disjoint subtours and then connect the subtours. This crossover proves to be meaningful and helps the algorithm avoid local optimal traps. Once again, the excellent behavior of GA-EAX on the CTSP instances is consistent with its performance on conventional TSP instances as shown in *Nagata & Kobayashi (2013)*.

## CONCLUSION

This work presents the first extensive computational study on the transformation approach of solving the Clustered Traveling Salesman Problem with general TSP solvers. Based on the results from the exact Concorde solver and the heuristic CLKH and GA-EAX solvers on 20 medium ($101 \leq |V| \leq 1,000$) and 15 large ($1,173 \leq |V| \leq 2,000$) CTSP benchmark instances and 38 large GTSP benchmark instances (with up to 24,978 vertices) available in the literature, we can draw the following conclusions.

- The exact Concorde solver can optimally solve all medium and large CTSP instances. It also solves exactly large GTSP instances with up to 3,162 vertices in a reasonable time, but fails to solve larger GTSP instances in 24 h. Its solution time is not completely consistent with the size of the problem instances.
- The heuristic CLKH and GA-EAX solvers perform very well both in terms of solution quality and computational efficiency. Both solvers have a good scalability, making them particularly suitable for solving very large instances with at least several thousands of vertices. For the tested instances with up to some 24,978 vertices, GA-EAX exhibits a better performance than CLKH.
- The general TSP solvers significantly dominate, both in terms of solution quality and computational efficiency, the current best performing CTSP heuristics specially designed for the problem. In particular, the TSP heuristics are several orders of faster than the state-of-the-art CTSP heuristics to find much better results.

This study indicates that the existing CTSP benchmark instances in the literature are not challenging for modern TSP solvers even if they remain difficult for the existing CTSP algorithms.

Finally, given the findings of this study, it would be interesting to investigate the problem transformation approach for solving other TSP variants that can be converted to the TSP or to a TSP variant for which effective algorithms are available.

# ACKNOWLEDGEMENTS

We are grateful to the Associated Editor and anonymous reviewers for their valuable suggestions and comments which helped us improve the paper.

## Funding

This work was supported by the National Natural Science Foundation of China (No. 72122006). The funders had no role in study design, data collection and analysis, decision to publish, or preparation of the manuscript.

## Grant Disclosures

The following grant information was disclosed by the authors:
National Natural Science Foundation of China: 72122006.

## Competing Interests

Jin-Kao Hao is an Academic Editor for PeerJ Computer Science.

## Author Contributions

- Yongliang Lu conceived and designed the experiments, performed the experiments, analyzed the data, performed the computation work, prepared figures and/or tables, authored or reviewed drafts of the paper, and approved the final draft.
- Jin-Kao Hao conceived and designed the experiments, analyzed the data, authored or reviewed drafts of the paper, and approved the final draft.
- Qinghua Wu conceived and designed the experiments, performed the experiments, analyzed the data, performed the computation work, prepared figures and/or tables, authored or reviewed drafts of the paper, and approved the final draft.

## Data Availability

All data and code are available at GitHub: https://github.com/lyldft/TSP.

## Supplemental Information

Supplemental information for this article can be found online at http://dx.doi.org/10.7717/peerj-cs.972#supplemental-information.

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
