# Peer review of "Solving the clustered traveling salesman problem via traveling salesman problem methods"

_PeerJ Computer Science, doi:10.7717/peerj-cs.972_

## Round 0.1 · original submission · Major Revisions

A revision is needed before further processing. Please address the concerns raised by the reviewers. Note that I do not expect you to cite any recommended reference unless it is essential. Thanks.

·

Basic reporting

In this paper, the authors investigate the clustered traveling salesman problem (CTSP), which is an extension of the popular TSP.

The paper is clear and the conclusions are supported by extensive computational results.

I do recommend to add at the references the following paper:

P.C. Pop, I. Kara and A. Horvat Marc, New Mathematical Models of the Generalized Vehicle Routing Problem and Extensions, Applied Mathematical Modelling, Elsevier, Vol. 36, Issue 1, pp. 97-107, 2012.

which presents some interesting MIP formulations of the CTSP, and the papers:

O. Cosma, P.C. Pop and L. Cosma, An effective hybrid genetic algorithm for solving the generalized traveling salesman problem, Lecture Notes in Computer Science, Vol. 12886, pp. 113-123, 2021.

P.C. Pop, O. Matei and C.M. Pintea, A two-level diploid genetic based algorithm for solving the family traveling salesman problem, in Proc. of GECCO 2018, Association for Computing Machinery, Kyoto, Japan, 2018.

These papers deal with problems closely related to CTSP: the generalized traveling salesman problem (GTSP) and the family traveling salesman problem (FTSP). It is important to mention these papers in the introduction, especially because the authors are using GTSP instances.

Experimental design

The authors used the transformation suggested by Chisman in order to transform the investigated problem into classical TSP, which then is solved using state-of-the-art algorithms.

The research questions are well defined and they are relevant.

Validity of the findings

The conclusions are well stated and they are clearly linked to the proposed research questions.

Therefore, taking into consideration all the mentioned aspects I do recommend to accept the paper for publication if the authors answer to the addressed comments.

Reviewer 2 ·

Basic reporting

This manuscript investigates methods for solving the Clustered Traveling Salesman Problem (CTSP), specifically, by converting it to the Traveling Salesman Problem (TSP) that has been extensively studied. The experimental results essentially answer several of the questions proposed by the authors.

Experimental design

no comment

Validity of the findings

no comment

Additional comments

There are several issues below that need further clarification by the authors:

1. In Section 1, in addition to introducing the literature involved in the relevant methods, the characteristics and scope of application of the corresponding methods should also be summarized.

2. Again in the Section 1, some references seem to be too old.

3. The focus of this paper is to explore the conversion of CTSP to TSP for solving. The conversion method adopted by the author and the considerations behind it should be described in more detail in Section 2.1.

4. On page 8, line 285, what does “Err” in “B-Err” stand for? Authors may consider using more appropriate abbreviations.

5. The author's research found that several TSP solvers have different performance for CTSP. However, it would also make sense to discuss the reasons behind these differences in further depth.

6. When comparing the results, some statistical methods can be added to compare the superiority of several methods, such as variance analysis, Friedman test, etc.

---

## Round 0.2 · accepted · Accept

The paper can be accepted now. Congratulations.

·

Basic reporting

The authors took into consideration all the comments and observations and as a consequence, I do recommend to accept the paper for publication.

Experimental design

No comment.

Validity of the findings

No comment.

Reviewer 2 ·

Basic reporting

most of the comments have been addressed, just need to do proofreading before publication

Experimental design

most of the comments have been addressed, just need to do proofreading before publication

Validity of the findings

most of the comments have been addressed, just need to do proofreading before publication